# Early Investigation of a Landslide Sliding Surface by HVSR and VES Geophysical Techniques Combined, a Case Study in Guarumales (Ecuador)

Olegario Alonso-Pandavenes [1,*] , Francisco Javier Torrijo [2] , Julio Garzón-Roca [3] and Alberto Gracia [4]

1 Geology and Mining Engineering Faculty-FIGEMPA, Central University of Ecuador, Quito 170521, Ecuador
2 Research Centre for Architecture, Heritage and Management for Sustainable Development (PEGASO), Department of Geotechnical Engineering, Universitat Politècnica de València, Camino de Vera s/n, 46022 Valencia, Spain
3 Department of Geodynamics (GEODESPAL), Faculty of Geology, Complutense University of Madrid, 28040 Madrid, Spain
4 Consultores Técnicos Asociados (CTA), S.A.P., 50006 Zaragoza, Spain
* Correspondence: omalonso@uce.edu.ec; Tel.: +593-995608066

**Featured Application: A new methodology in the application of the HVSR technique using VES geoelectrical surveys combined for landslide studies.**

**Abstract:** The access road to the powerhouse's hydraulic system's facilities in Guarumales (Azuay, Ecuador) presents a medium-sized landslide. Geophysical tests were conducted in the initial research stage, combining electrical and seismic methods. A vertical electrical sounding (VES) and horizontal to vertical spectral ratio (HVSR) survey campaign have been taken as a reference for the analysis of the landslide area. The distribution of these test points has been at three different levels along the landslide where the access road crosses it, trying to cover the area's most extensive possible length and width. In the area, we find the geology dominated by the presence of schists, altered to different degrees and presenting blocks of material with a lower degree of alteration within colluvium formed by a clayey matrix and coarse material of the exact nature. There is also observed runoff water and groundwater in the area. The results obtained through SEV tests have allowed for defining the separation zone of the mobilized or sliding materials compared to the fixed or immobile ones (potentially, the sliding surface was marked). Using the HVSR technique, the natural vibration frequencies of the ground associated with the sliding mass (separation of seismic impedances between a two-layer model: mobile and fixed) have been determined. Previous authors proposed an empirical relationship establishing the exponential relationship, already proposed by previous authors, between sediment thickness and natural frequencies. It has been possible to determine the depth of the position of the loosely compacted sediment zone (and probably moving or mobilized) compared to that of compact materials (immobile) and thereby define the potential rupture surface.

**Keywords:** Guarumales (Ecuador); landslide; HVSR; VES; rupture surface

## 1. Introduction

The southeastern part of Ecuador is prone to landslide events, especially those related to human activity, such as the access roads construction in areas with steep slopes. These actions and constructions are among the most important triggers in geohazard landslide occurrences that produce high economic losses and even affect lives [1]. Thus, understanding its moving processes and identifying and mapping the rupture line is one of the initial studies related to performing optimized applications of investigation surveys conducted to eliminate or minimize them [2].

A comprehensive kind of data could be collected by applying superficial surveys and studying subsurface conditions, most of which use bore-hole probes (perforation,

logging, or inclinometer tests) as a direct source of an information database. However, the transportation and location of drill machines and their application processes in that high-slope or soft-soil areas (sometimes both) involve expensive economic outlays. Also expensive is the use of UAV photogrammetry or A-DInSAR interferometric methods [3]. It is hard work, and most are challenging to perform, even create or increment mass movements. Geophysical surveys are a low-cost tool and an easy way to perform deep investigations and get valuable information in a short time that can be correlated with geological or geotechnical parameters [3,4].

The application of shallow geophysical investigation in landslide studies is traditionally related to: (i) seismic methods, such as refraction and passive techniques, from the investigation of the transmission velocity of elastic waves ($V_p$ and $V_s$), and provide information on materials' characteristics, geometry, and 2D distribution and can yield some geotechnical-related parameters such a density, compaction, and elastic dynamic modulus; active and passive techniques can define the whole geology, the differentiation between sediments and the basement, and even the sliding surface [3–5]; (ii) electrical methods, such as Vertical Electrical Soundings (VES) and tomography, are the most employed geophysical surveys in soft soils landslides investigation; they are used to define the position of the phreatic levels (humidity and presence of aquifers levels) and also to draw its internal structure, faults and fractures, and stratigraphic relationships; research related to the measurement of resistivity changes allows for defining the area's stratigraphy (separation of material layers) and the presence of humidity and water (even degree of saturation) in the geological levels. Furthermore, 2D techniques such as tomography allow for obtaining and defining structures (faults and joints) and geometric relationships between materials [4–6]; (iii) others include electromagnetic (EM) techniques such as Ground Penetrating Radar (GPR) [3–6]. The GPR is important for studying the shallow characteristics of materials, but its capability to get some in-depth information is limited [3–6]. The combination of these techniques, involving direct investigations as dill holes, makes it possible to define and study the geometry of a landslide and, on occasion, to establish its rupture surface [5,6].

Moreover, passive seismic techniques have recently begun to be broadly applied in ground investigation, such as multispectral seismic and spectral analysis of components and are combined with other methods transported by drones (UAV) or even interferometric techniques [3]. Therefore, Multichannel Analysis of Surface Waves (MASW) and Horizontal to Vertical Spectral Ratio (HVSR) techniques have started to be more useful in landslide studies [4,7–13].

The research objective has been to study an area in the process of landslide in an initial phase, which involves determining a fast and effective enough amount of information that allows for both the proposal of methods of remediation or mitigation in this phenomenon. This early geophysical campaign was designed to obtain previous knowledge about the size and depth of this landslide in order to study it entirely and in in-depth in the future. It will be the basis for defining where and how deep drilling bore-holes can be.

This initial data could be capable of establishing advanced research criteria for the landslide, such as the location of drill holes or even a new advanced geophysical campaign. The geophysical research investigation proposed must be reliable, easy to perform, and give some accuracy in interpretation models.

The methodology used in this research combined the application of two geophysical methods: seismic and geoelectrical. Refraction and MASW techniques were applied (as active profile ones), and the HVSR passive technique from the first method was used as was the VES technique of the geoelectrical method.

The results obtained from applying refraction and MASW seismic techniques will allow for obtaining velocity values of the compressional (P) and shear (S) waves, respectively, which will be used in defining the characteristics of shallow and probably moving materials [5,14]. It also can provide a two-dimensional view of the geometry of shallow layers and identify the separation between soft materials (moving sediments) and hard rock (considered fixed). On the other hand, the VES results could define the thicknesses

of the altered materials overlaying the rocky material related to the static basement (now related to resistivity) and identify the presence of humidity in materials; therefore, it will complement the active seismic profile results [4].

The HVSR single station survey will provide the fundamental vibration frequency ($f_o$) as the principal value and will be correlated to sedimentary thickness overlaying the rocky basement [15–17]. Some of them will be performed at the same VES test points, which can be defined as control points to establish an empirical relationship.

That empirical relationship could be determined for the whole area between said thicknesses of the mobilized materials (sediments) and the fundamental frequency of ground vibration obtained in the HVSR surveys [18].

Different authors have established and tested this application relating to the thickness of sediments over a basement obtained in drilling surveys with HVSR measurements [10,18–20]. However, other geophysical techniques, such as VES, still need to be developed as empirical references. Only authors such as [21] have established this kind of relationship between the thicknesses of the loose materials on a sedimentary basin obtained from VES surveys and the HVSR frequencies with good results.

The results of this investigation have made it possible to establish the position of the thickness sediments under every HVSR test. Thus, support this methodology aims to extend the results at various points in the study area only by applying a punctual device (HVSR single station measures) through a simple and economical procedure. Therefore, it can provide an overview of the sedimentary material overlaying compact material and correlate it with the rupture surface's position.

## 2. Background and Scope

### 2.1. Geographical Situation

In an area near Guarumales county (Azuay province, Ecuador), the CELEC Public Company South-Demarcation has an unpaved road that connects the main national route E-40 with the facilities buildings. It is the only way to access the powerhouse of the Sopladora Hydroelectric Power Plant (SHPP).

This road has a sinuous S-shaped descending path along a significant slope from an elevation of 1780 m above sea level (m.a.s.l.) at the top intersection point and with the buildings, facilities, and installations located at an elevation of approximately 1020 m.a.s.l, at the same level of the Paute river course.

The area is close to the community and population of Amaluza (central coordinates UTM-datum WGS-84 zone 17M: 784,104 E/9,711,897 N), where all surrounding landscapes have pronounced slopes. It is a complex geomorphology area dominated by V-shape snaked-type valleys with slopes sides of more than 45° on average (Figure 1).

No previous investigations were conducted in the area by owners, and only a mitigation action was applied to the damaged road by the slide of materials that affected the circulation of people and vehicles to the facilities area (Figure 2).

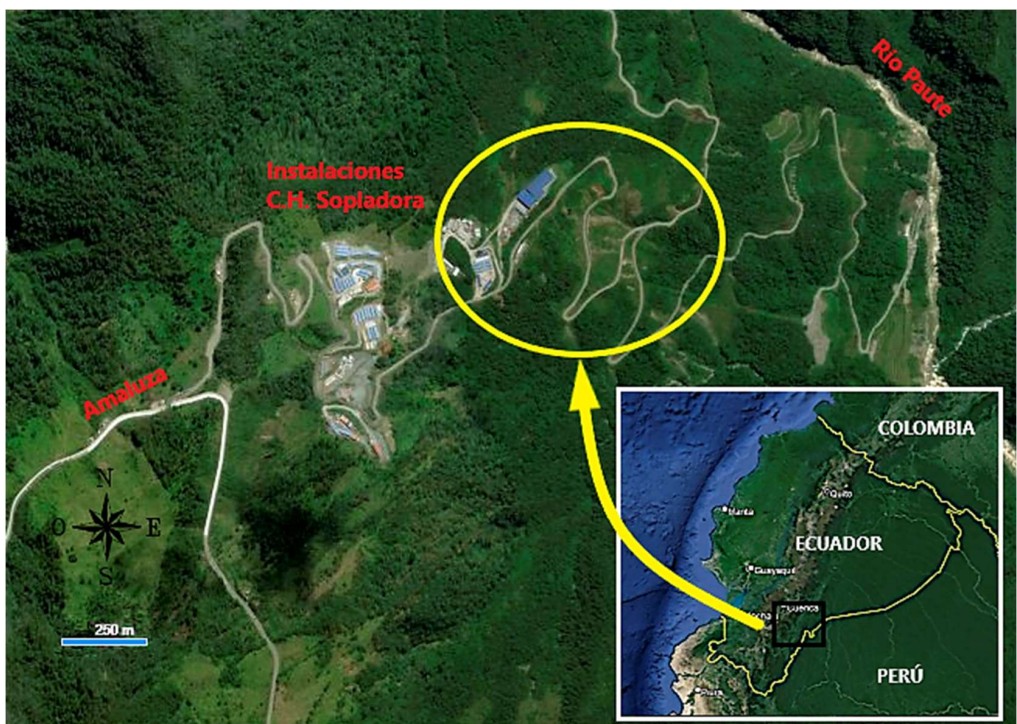

**Figure 1.** The situation of the investigation area. The yellow circle locates the sliding area (Modified from [22,23]).

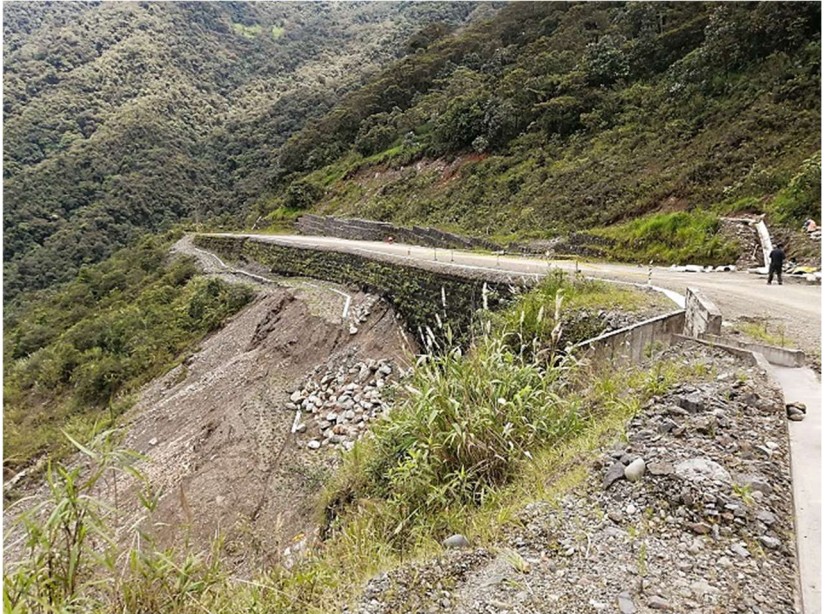

**Figure 2.** A view of the middle part of the landslide area where the road was affected (cut) and reconstructed, applying observed earth-walls. In this location, the movement is prone to continue at a low displacement tax. The high slope on the landscape can be observed.

## 2.2. Geophysical Background and Scope

The use of geophysical research in landslide studies or the excavation of slopes is widely recognized, with seismic and electrical methods and their techniques being the more commonly used in these investigations [4–6,24]. The sliding mass and its movement generate changes in measured geophysical parameters, so it can be used this kind of surveys

to analyze this internal ground modifications and recognize the landslide body or even study its dynamic motion [5].

The HVSR passive seismic technique has been defined and developed by Nakamura [15] in order to establish the fundamental ground frequency ($f_o$) values for a two-layer model (soil and compact material or basement). In this proposed model, the fundamental frequency is related to the soft or sedimentary layer shear-waver velocity and its thickness [15,16]. Therefore, if that shear wave velocity ($V_s$) can be estimated or measured, it can be calculated and defined as the soft overburden material that lies over the basement.

That assumption is valid on homogeneous media such as a sedimentary basin or a broad soil-rock system where vs. changes are not so significant. However, it can produce some errors if the soil or sediments have lateral changes in composition and/or compaction. There, this relationship causes irregularities in the definition of the depth of the basement [17].

HVSR surveys are based on the knowledge that seismic noise in the indicated model is composed of surface waves and media with enough impedance contrast (related to the density and seismic velocity). Therefore, complex phenomena can be produced where the soft sediments have changes in their density and/or vs. (heterogeneities), and results have no accuracy [25,26].

In some landslide material conditions, where a high contrast impedance could exist (soft soils or sediments directly over a rocky substrate, as in this investigation area), shear and compressional waves can predict its position [27]. Nevertheless, compacity or density variations could still affect the relationship, and more than the HVSR technique is needed [28,29].

HVSR is a recognized technique to define $f_o$ from a sedimentary layer over a compact material [30]. However, it has limitations for other purposes such as vs. determination [31]. Recently, the possibility of its use for other purposes and different methodologies has been investigated.

The research of authors such as [18,32] in establishing empirical correlations that relate the thickness of the surface materials (soils and soft sediments) with the value of said fundamental frequency $f_o$ was considered This relationship is expressed from a potential equation of the type:

$$Z = a\, f_o^{-b} \tag{1}$$

where $Z$ would be the depth at which the rocky substrate is found (or what is the same, the thickness of the sediments that underlie it), and $f_o$ would be the value of the fundamental frequency of the ground measured in the HVSR survey. Parameters "$a$" and "$b$" would be factors to be defined experimentally, which are related to the nature of the materials and the research area [33].

Most of the investigations about this formulation have been based on the use of mechanical drill holes to define the thickness of materials that lie over a rocky basement [19,20,34,35]. This type of correlation is considered to have a high accuracy in establishing the relationship when using a direct prospecting method.

These applications of the HVSR technique have been increasing in recent years with landslide investigations, almost always combined with drilling or other geophysical investigations such as refraction seismic and electrical tomography [7,36–39].

In a few cases [4,6], other geoelectrical surveys have been used as a reference in investigations, such as VES tests in [21], where they only apply this test type to establish the indicated correlation.

In the present investigation, two different combined methods have been used. The seismic method was applied through active seismic techniques such as refraction and MASW profile techniques and HVSR seismic passive surveys. Complementarily, the geoelectrical method was proposed using the technique of VES.

The VES technique has been widely used and developed over time regarding the geoelectrical method. The generation of an electric field and the measurement of its potential variation when crossing the ground allows for obtaining resistivity values that,

depending on the configuration of the electrodes, can reach depths of several meters and is one of the most used geoelectric techniques in the investigation of landslides, along with electrical tomography [4].

In the seismic method, the seismic refraction technique allows, depending on the arrival time of an elastic wavefront to the receivers arranged along a profile (geophones), for determining the compression velocity $V_p$ and the possible variations and changes in the composition and geometric configuration of the materials. The relatively simple test has extensive literature and knowledge and can be applied to mobilized landslide materials [3].

On the other hand, the surface waves analysis surveys of the seismic method, such as the MASW, allow, from the frequency analysis of surface and body waves, to obtain the distribution of values of the velocity of the shear wave S ($V_s$) in the analysis of a linear system of geophones with a one-dimensional distribution [40,41]. The shear wave velocity in the ground usually complements the data obtained in the refraction seismic.

The use of VES surveys as a correlation tool will be proposed without the need to carry out bore-holes (mechanical drilling) with the consequent reduction in costs and speed of execution of the study. For this, the variation between resistivity will be used as a differentiation tool. The shallow materials (less compact) and potentially in movement (sliding) present a contrast against the static rocky substrate (and of greater resistivity). Here, complementary tests will be available, such as active and passive seismic tests for model calibration.

The combination of geophysical techniques, as [42] resumes, complemented with direct surveys, such as drilling, allows for determining the first parameters in landslides. Moreover, they can define the position of the rupture surfaces with considerable precision, always depending on the conditions and characteristics of the area and materials [4,6,28,43].

Therefore, in order to study, in advance, the sliding zone (the material that was in movement) and define its thickness, a geophysical research campaign application has been proposed. The company scopes are to analyze different kinds of materials, obtain some knowledge about the presence of water in materials, and define the landslide rupture surface. This information could provide an early idea of the size of the landslide, materials involved in the sliding area, an approach to the focused areas, and the investigation depth in future works.

These investigations will be the basis for decision-making in future actions regarding landslide research through mechanical soundings and monitoring of the event using inclinometers.

## 3. Geological Knowledge

Ecuador has complex geology as a product of the close subduction area where the Nazca plate goes down the South American one. That is the origin of the Andean mountains and the volcanoes chain with a north to south general direction. Because of this process, five litho-tectonics and geologic trends can be defined almost parallel to the Pacific coast. The southern part of the country suffered high-pressure tensions, and the old sedimentary basement transformed into metamorphic rocks [44].

The investigation area is at this regional metamorphism belt in the oriental part of the Andes mountains. The local geology is characterized by outcrops of metamorphic materials from the Upano Unit, belonging to the Salado Group, which Litherland had defined in 1988. This metamorphic basement is composed of: metandesites, green and sericitic schists, tuffs, and metagrauwakes that occupy a vast expanse of land throughout the area that includes a strip of about 30 km wide and more than 60 km long, with a north–south direction, approximately. These basement materials in the study area have layer directions against the slope, and the stratigraphy may present dip angles up to 70° to the west in some areas (Figure 3).

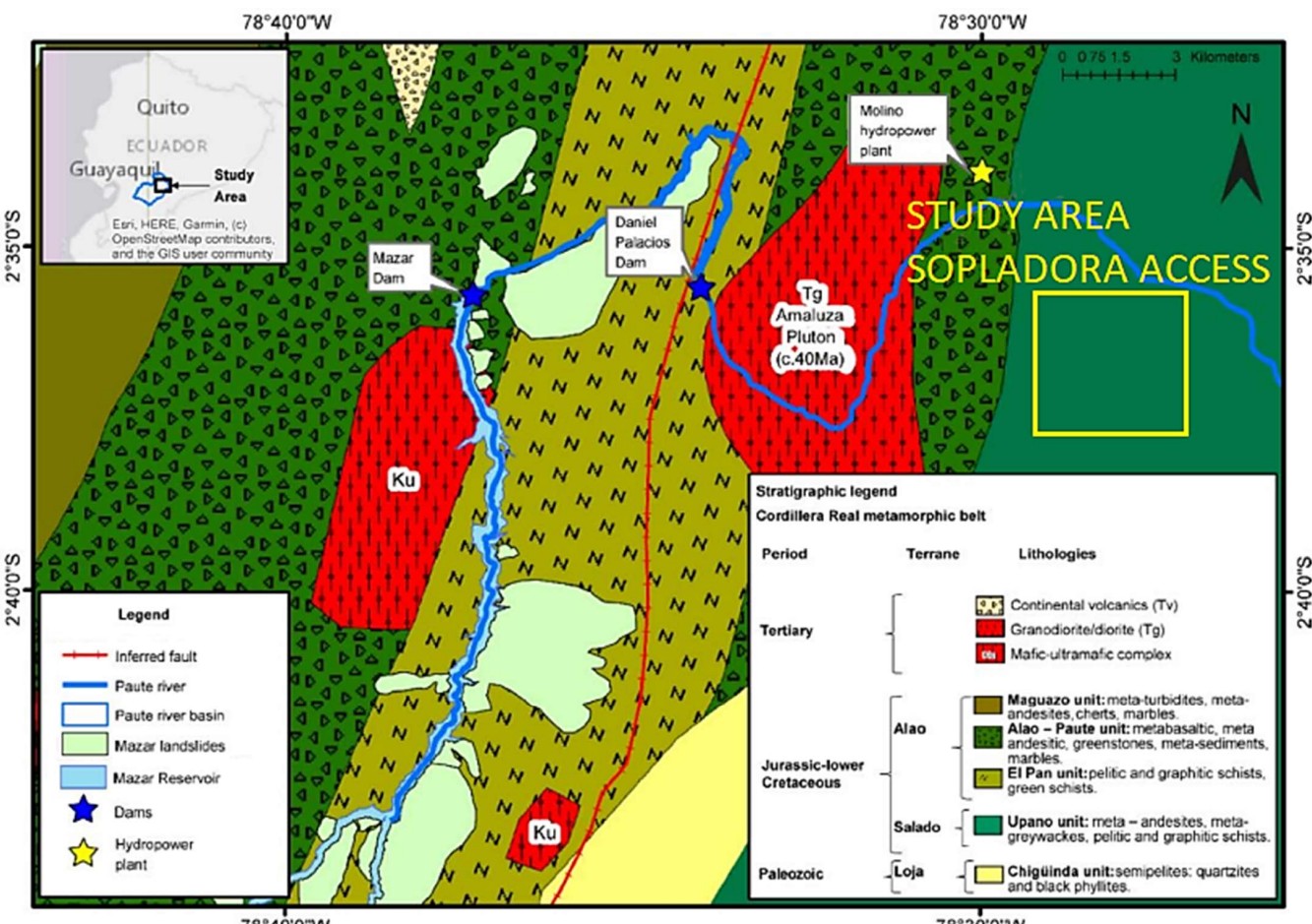

**Figure 3.** A regional geological map with the study area (the yellow square) showing the general trend of geologic materials and present formations. Modified from [44].

In the investigation area, the graphitic schistose basement is covered by surface materials where a part may be eluvium. Various degrees of metamorphic basement alteration has been observed, which at higher levels become clay-type soils of alteration [44,45]. In the work area, it has been possible to observe the presence of degrees of alteration IV-V (as residual soils) forming part of the materials in contact with the rocky substratum. Up to more than 10 m of power of this degree of alteration has been evidenced [44].

This shallow residual soil, formed from the severe alteration of the graphitic schists, is covered by transported sediments in the aqueous phase (most fine-grained) or by gravity (colluvium with coarse grain sizes) with a centimetric level of organic top-soil. The colluvium originates because of the steep slopes observed in the area and the high rainfall, which weathered the substratum and generated low cohesion between silt and clayed materials [44]. The presence of colluvium, as a product of ancient small-magnitude landslides and gravitational sedimentation, with varying thicknesses (from 3 to 5 m thick up to >10 m), has been determined and observed in the field. In both cases, the eluvium and colluvium layers present a matrix composed of clay-loam-type materials with coarse material inside, such as pebbles and gravel. It also contains medium to large-sized blocks (diameters greater than 1.0 m), which present minor degrees of alteration (Grades IV-III) and lie at the bottom of this sediment layer (see Figure 2).

The area's geomorphology is that of a somewhat open V-shaped valley with general slopes of the flanks ranging from 35° to 45° and that locally can reach an over 60° incline (in rocky substrate outcrops). The bottom of this valley is crossed by the Paute river from the northwest to southeast in the northeast study area (see Figures 1 and 3).

The landslide-studied slope has an inclination to the east with an average of approximately 40° to 45°. Three different positions in the slope have been cut by the constructed section of the affected access road (see Figures 1 and 2). The shallow materials where the geophysical surveys were applied are clayey silt and silty clay with pebbles and blocks (30% in proportion to the matrix) that have been superficially reworked by a runoff action of water and covered by artificial fills from construction roads.

## 4. Geophysical Research and Applied Methodology

### 4.1. Geophysical Surveys

Geophysical research has applied two different geophysical methods combined: distributed geoelectric and seismic, as shown in Figure 4. The designed campaign of surveys and applied techniques have been reduced to the available spaces in the studied area. They were the access road over the three levels where it cuts the slope (Figure 4): one at the center of the landslide and the other at both sides (upper and lower zones). It was because the slope of the area being studied is very steep and has thick material on the surface (see Figure 2). Little compact shallow sediments are prone to movements or punctual landslides, and the falling of loose material could interfere with data quality acquisition or prevent the correct application of the measuring devices (even personal security). That is also an obstacle to installing drill hole machines so that the results can provide better locations in further studies.

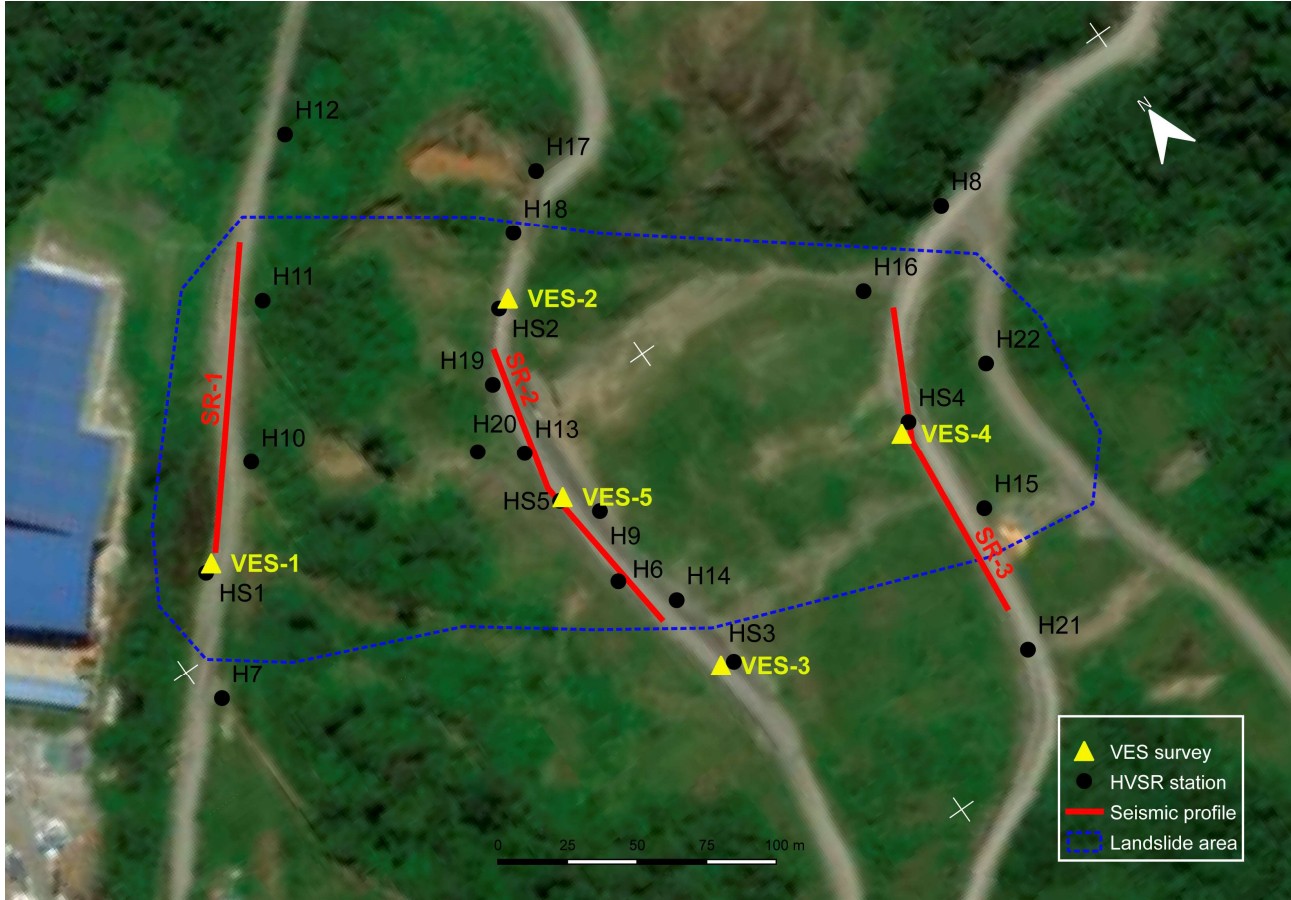

**Figure 4.** Situation map of surveys (noted that the north is rotated counterclockwise) and observed landslide area (blue dashed line). Black dots: HVSR surveys; yellow triangles: VES surveys; and red lines: seismic refraction and MASW profiles, both made at the same position. White crosses refer to the coordinate points (Modified from [22]).

Concerning the geoelectrical method, the VES technique has been applied in five test points (yellow triangles in Figure 4). These surveys use an electrical field to define the resistivity of materials into the ground. The electric current was injected through the terrain from two electrodes (A and B), whereas the potential was measured in the other two electrodes (M and N). The use of different geometries in the A-M-N-B disposition is called measurement arrays, and they can be seen in different ways (as Wenner, Schlumberger, Dipole, for example) [4]. VES surveys use a 400 m-length AB electrodes aperture applying a Schlumberger array, providing information on materials and stratigraphy about the first 60–80 m from the surface elevation application point. The performed interpretation could show the presence of water and altered materials over the basement. In addition, it can be the tool that clearly defines the basement position from the resistivity contrast between rock and shallow sediments and identifies the stratigraphic levels.

The seismic method has been applied through two different techniques: active refraction and type MASW profiling (shown as red lines in Figure 4). Both techniques were performed over the same line to provide information about S-wave and P-wave velocities so they can be correlated. Using these seismic techniques can define the substratum interface as a high increase of velocities and complementary show variations in the compaction of materials [14].

The three refraction seismic profiles all have a 115 m length with a 24-channel array and five different shooting points. They have been executed at three levels on the road path of the access road from the upper intermediate to the lower part.

The first one, named SR-1, was performed at the high elevation area of the landslide, whereas SR-2 was at the middle elevation and SR-3 on the lower part, near the supposed landslide toe. The final model can obtain more accuracy and resolution by combining with VES interpretation.

Another technique was the punctual or single station passive-type HVSR, applied over 22 station points (represented in Figure 4 by black dots). This technique is easy to apply, and it was acquired by 3-direction geophones equipment with a 2.0 Hz frequency. It used a 20 min time record to have enough information from every single station. The results from data processing define the fundamental frequency of vibration ($f_o$) of the ground, which is related to thickness [15]. Complementarily, it can use the spectral ratio ($H/V$) or amplification ($A_o$) in further calculations.

When used as control points, five of the 22 HVSR surveys were executed at the same position as the VES tests (named differentiated with an S before their order number in Figure 4). A summary of the tests carried out can be seen in Table 1.

**Table 1.** Resume of applied geophysical techniques and surveys.

| Geophysical Method | Survey Technique | Dimension | Number of Surveys | Parameter |
|---|---|---|---|---|
| REFRACTION | 2D profile | 115 m | 3 | $V_p$ |
| MASW | 1D profile | 115 m | 3 | $V_s$ |
| VES | Application point | 400 m (A-B) | 5 | Resistivity |
| HVSR | Single station | Point | 22 | $f_o, A_o$ |

### 4.2. Applied Methodology

The methodology proposed in this investigation consists of the correlation of the geophysical data obtained between the HVSR surveys and the techniques of refraction seismic, MASW seismic, and VES. That will define an empirical relationship between fundamental frequency and sediment thickness. Equation (1) can provide an easy way to delineate the sedimentary ground (related to mobilized landslide material) over the fixed rocky substratum, i.e., the rupture surface.

The first step is to analyze the shallow material and define the sedimentary layer (even the altered material) overlaying the rock basement. Once the values of the thicknesses

of the most superficial materials are obtained, they could be related to altered ones and, consequently, potentially moving layers.

Thus, the position of the rocky substrate will be identified through changes in resistivity (by electrical method) and seismic impedance (by seismic method). That simple geological model (defining the soft layer thickness) was related to the natural frequency of vibration ($f_o$) obtained from the HVSR survey from a two-layer model [15], and it could be established as an empirical relationship between both values [12,13]. From this relationship, $f_o$, as a tool to delineate the depth of the basement, can be used [18–21].

Firstly, the apparent resistivity field curves obtained in the VES surveys have been inverted, and a layers' vertical distribution (resistivity and thickness) was performed for every test. These results provide the first definition between the rocky basement and sedimentary overburden from a high contrast in resistivity values.

The second part consisted of processing the seismic data of refraction and MASW surveys with obtaining distribution profiles of the values of the velocities $V_p$ (two-dimension profile with the geometry of geophysical layers) and vs. (one-dimension distribution of velocity and depth). Compared with the resistivity changes mentioned above, these values allow for obtaining the position of the rocky substrate in-depth (more accurate), that is, the thickness of the sediments and differentiation of internal levels in the overburden.

At every point where VES tests have been carried out, an HVSR test has been applied. These combinations of HVSR and VES results were used to establish the empirical correlation between sedimentary thickness with the fundamental frequency of the ground, and they are used as control points [18–21].

Once establishing this relationship, its graphic representation can be adjusted to a potential type curve based on Equation (1), in which all the parameters that will be defined are considered. Therefore, the $f_o$ values will be related to the thickness of sediments obtained at these control points [18–20]. This equation will make it possible to calculate the depth of the rocky substrate under the rest of the HVSR measurement stations and thereby draw an isopach map of sediment thickness and the definition of the slip rupture plane.

From the geological point of view of this slide area, the separation surface between the shallow materials (sedimentary or altered) and the rocky substrate will be established as the position of the landslide rupture surface. It can be defined because of the geological materials' conditions and their direction and dip indicated in the previous paragraphs.

Finally, the parameter established by [15], called the Vulnerability Index ($K_g$), will also be analyzed, which is related to effective stress, that is, the probability that materials have the potential to move or change their tensional conditions.

In the same table are also included the results of the calculation of the Vulnerability Index value, the $K_g$ parameter from [46]. These values are obtained from the following equation:

$$K_g = \frac{A_o^2}{f_o} \tag{2}$$

$K_g$ has a dimension of period (seconds) but is considered dimensionless, since $A_o$ is dimensionless.

## 5. Results

The interpretation of the different surveys carried out in the studied area has allowed for a sufficiently precise general definition of shallow sediments layered over the metamorphic substratum. It can provide the correlation between the applied different techniques. Next, each of the techniques applied and their results will be presented.

### 5.1. Seismic Refraction Technique

Results obtained correspond to the identification of five geophysical levels as a function of P-wave velocity values, except for SR-1, which only has three. The last level in all profiles corresponds to the semi-infinite space (Figure 5).

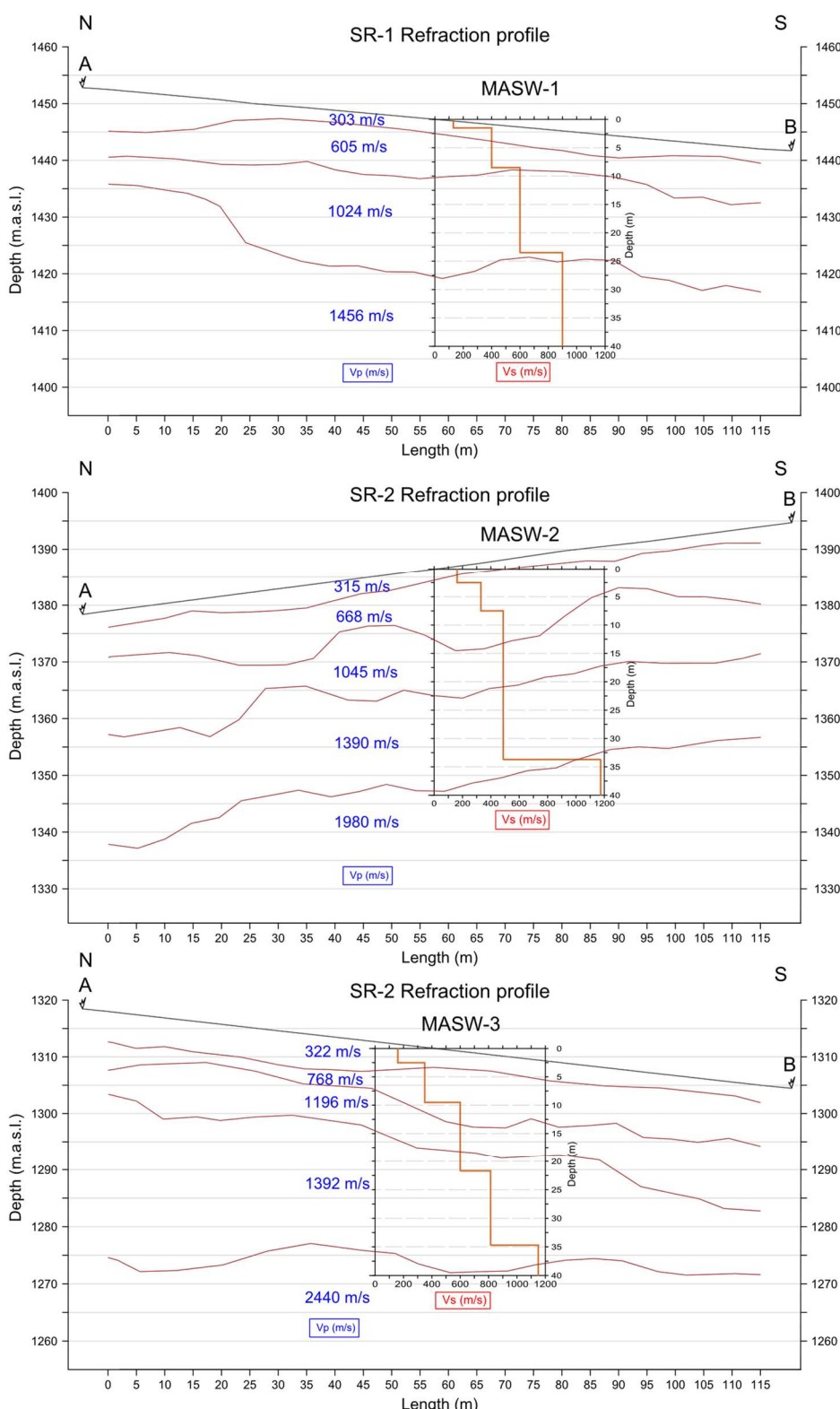

**Figure 5.** Two-dimensional refraction seismic interpretation profiles with $V_p$ values (in blue) and overlayed MASW profiles results ($V_s$ in red squared lines, 1D solution). Both profiles were made at the exact location.

The geophysical levels obtained are correlated with the area containing the observed geological materials and described from the top to the depth for all profiles. The first level, the shallowest one, corresponds to low-compact natural fill materials ($V_p$ values obtained

are between 303 and 322 m.s$^{-1}$) and could be composed of some artificial fills used for road construction. Below this level, more compact materials appear with P-wave velocities between 605 and 768 m.s$^{-1}$ related to more coarse sediments. Both levels are correlated to the gravitational sediments observed around the soils and colluvium.

The next level, the third one, is defined between 1024 and 1196 m.s$^{-1}$ P-wave values and would correspond to a more compact sediment level with boulders and/or rock blocks where probable humidity increases (Figure 5). The presence of a transition zone from these fillings to eluvial material could be correlated to the fourth level, where the compressional velocity is clearly over 1000 m.s$^{-1}$ (1390 to 1450 m.s$^{-1}$). No more interfaces were defined under this level at the SR-1 profile, so it is a semi-infinite space here.

The last geophysical level determined In the investigation (for SR-2 and SR-3, in Figure 5) has velocities of 1456 to 2440 m.s$^{-1}$, correlated with the basement's metamorphic substratum (this is the semi-infinite level). The impedance contrast values for this $V_p$ velocities between sediments and substrate varies from 2.2 to 2.3.

These materials' geometric distribution is undulated, showing a deeper thickness to the center of the profiles that turns thin at the sides. Its related P-velocities aim to determine an initial distribution model of geophysical levels in the area and provide an initial idea about the possible position of the landslide rupture surface at the sediments-basement interface separation (see Figure 5).

### 5.2. MASW Seismic Technique

MASW-type seismic profiles have been executed over the same position as the refraction ones to compare both interpretation models. In this case, the results are not in two dimensions (section-like) but a single dimension, thus obtaining only thicknesses of materials and shear wave velocities [40]. As a profile-style survey, the investigation involves all extensions where geophones are distributed, but the final values are assigned to the central point of each profile.

The results, in the distribution of layers of geophysical levels, can be correlated to those observed in the interpretation of the previous profiles (refraction ones), but now obtaining the S-shear wave velocity distribution model.

The results were represented in the same figure to show the comparison between both techniques: refraction and MASW (Figure 5). Here, the vs. average values obtained are a velocity of 142 m.s$^{-1}$ for the first layer, 318 m.s$^{-1}$ in the second, 581 m.s$^{-1}$ in the third, and 1020 m.s$^{-1}$ in the layer corresponding to the semi-infinite space, the last one. The correlation of these velocity levels with the area materials is the same as was described in the previous paragraph.

The MASW results corroborate the identification of the geophysical model of the studied area by refraction technique with a little difference in thickness that can be observed in Figure 5 (as a consequence of the analysis way that uses the techniques). The relationship between $V_p$ and vs. could define the dynamic elastic modulus and the impedance contrast in vs. velocities varies from 2.3 to 2.5 in the three analyzed profiles.

### 5.3. HVSR Seismic Passive Technique

The processing of the HVSR data has been carried out using the free software GEOPSY (www.geopsy.org, accessed on 10 January 2022), composing the records of the horizontal geophones (directions N-S and E-W) geometrically and applying the fast Fourier transform (FFT) on the time windows of 20 s established in the records once filtered [32].

The analysis in the domain of the frequency of the spectral ratio $H/V$ offers a result of a curve called ellipticity and is related to the surface waves, Rayleigh. In these curves, the maximum value or peak for the dominant frequency ($f_o$) is analyzed, which has an associated amplification $A_o$ of the signal (spectral ratio $H/V$).

The processing flow starts with filtering raw data to eliminate the transients. The time window size, established in 20 s in this case, and its selection allows the data to be computed in each geophone record. Then, FFT was applied to every selected window

before the geometrical combination of horizontal data. The GEOPSY software computes the H/V ratio and shows the spectral analysis in a frequency vs. H/V ratio (also called amplification of the signal, $A_0$) [32,33].

A single peak is usually shown in ellipticity curves, but some broad or multiple peaks can be obtained as well from processing. The graphics' different forms have the meaning of some basemen geometry variations (an inclination, for example), as the SESAME project indicates [32].

Figure 6 shows selected examples of HVSR processing results. The ellipticity curve in continuous black lines and its standard deviation (dashed black lines) are shown. The characteristic peaks for the $f_0$ (in hertz) value are indicated at the center of the two gray bars of different shades (these bars show its standard deviation). The first two graphics are from control points (HS1 and HS3), and the other four are from different elevation points, used as examples of the other sixteen surveyed points.

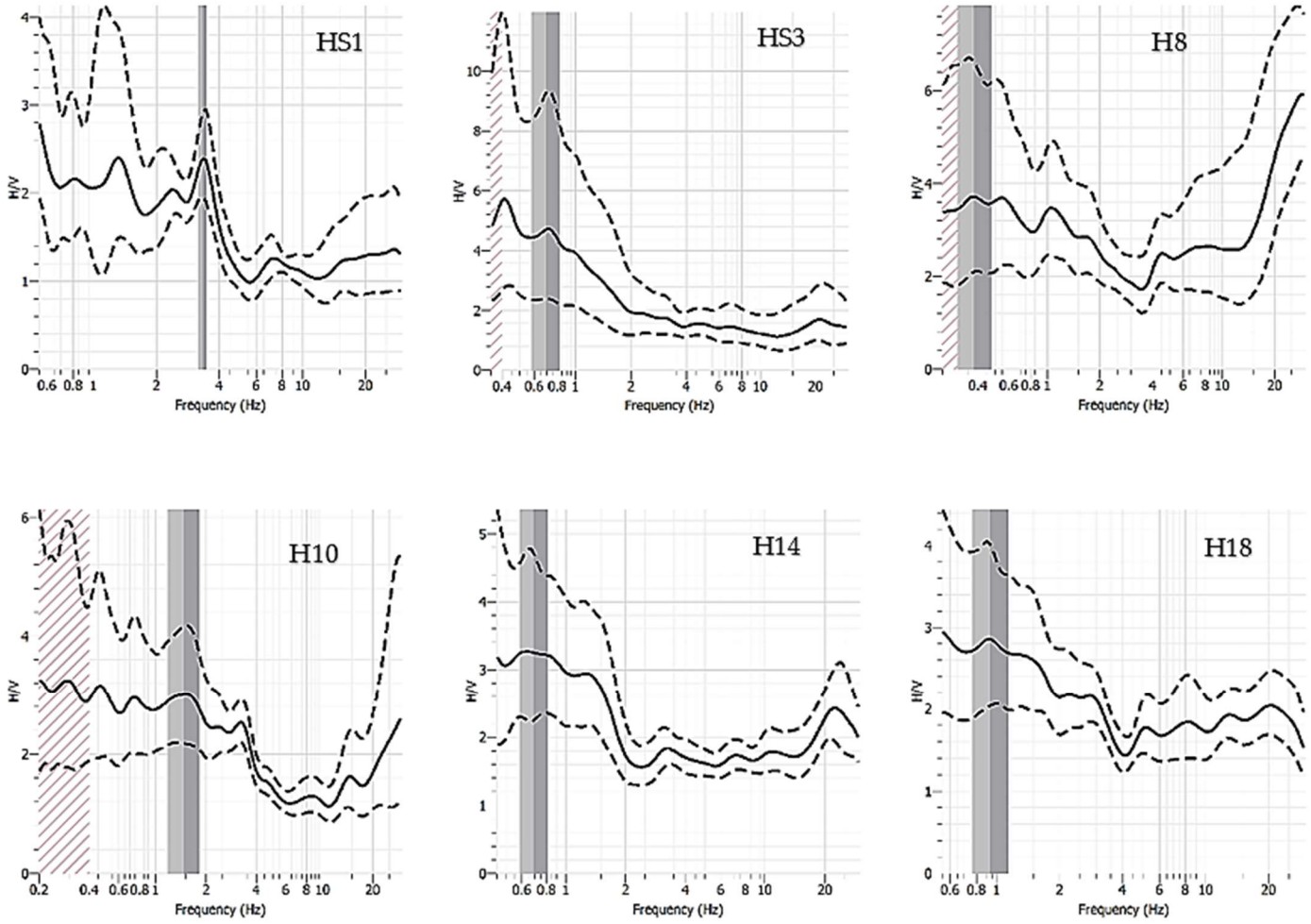

**Figure 6.** Example of six of the HRSV surveys processing, belonging to the proposed investigation. The two first are measured on control points (the same position as VES surveys). The gray vertical bars indicate the position of the fundamental frequency of each point, and the black dashed line shows the variation on the dispersion curve (continuous).

Results obtained in the HVSR technique single-station applied surveys are those shown in Table 2. The value of $A_0$ is dimensionless since it represents the spectral ratio $H/V$ of the horizontal components versus the vertical. The first five values, indicated with an S as a numbering prefix, would correspond to the parametric tests (performed in the same position as the VES surveys) to be used in the subsequent establishment of the empirical relationship.

**Table 2.** Results obtained in the interpretation of the HVSR tests ($f_o$ Hertz, $A_o$ dimensionless).

| HVSR Point | $f_o$ (Hz) | $A_o$ (Adim) |
|:---:|:---:|:---:|
| S1 | 3.21 | 2.37 |
| S2 | 2.61 | 2.05 |
| S3 | 0.70 | 4.70 |
| S4 | 0.49 | 3.42 |
| S5 | 0.25 | 3.91 |
| 6 | 1.07 | 4.19 |
| 7 | 3.66 | 2.28 |
| 8 | 57.67 | 12.66 |
| 9 | 0.34 | 6.49 |
| 10 | 3.28 | 2.53 |
| 11 | 2.42 | 1.90 |
| 12 | 3.76 | 1.54 |
| 13 | 0.73 | 3.18 |
| 14 | 0.60 | 2.99 |
| 15 | 9.37 | 2.34 |
| 16 | 1.04 | 2.99 |
| 17 | 4.81 | 2.02 |
| 18 | 2.95 | 2.13 |
| 19 | 0.71 | 3.42 |
| 20 | 0.91 | 3.22 |
| 21 | 25.30 | 2.38 |
| 22 | 1.18 | 3.17 |

Most of the graphical results show a broad peak type related to the SESAME project explanation of a high-angle inclination of the basement that could be present in this area. Otherwise, single, cleared peaks (as shown in HS1 point, see Figure 6) were obtained in several points.

The $f_o$ obtained values are comprised between the 57.67 and 0.25 Hz, with a 1.83 Hz average value (if we consider three higher values are excepted). For the amplification $A_o$, the obtained values in the entire performed surveys are between 12.6 and 1.54 (see Table 2). As a simple explanation, a higher frequency value is related to a shallow position of the rocky basement, whereas lower values indicate a deeper compacted substratum [32].

*5.4. VES Geoelectrical Surveys*

The VES surveys provided a distribution of materials that have been summarized in Figure 7. In the models obtained, the presence of a low resistivity level (of the order of 100 Ohm.m, marked with blue rectangles) is observed on materials that present a certain degree of alteration (example: VES-4 and VES-1) or on the ground without alteration (example: VES-2 and VES-3). A special situation is produced in VES-5 where alteration resistivity values reached more than 77 m under this low resistivity level.

Above this level of low resistivity, related to the presence of alteration clays and clayey silts with high humidity (or possible saturation), colluvium-type materials are found (presenting intermediate resistivity wide-range values), and are marked in Figure 7.

The high values of resistivity (>5000 Ohm.m) would be correlated with the unaltered rocky substrate (fixed material or static). Therefore, it is possible to proceed to the separation of the thicknesses of superficial sediment of lower electrical resistivity, which coincide with the possible presence of water (or high humidity) and/or clay materials (colluvium and alteration of schists).

Therefore, a correlation could be established between the position of the landslide failure surface and these obtained values of sediment thickness in the interpretation of the VES tests.

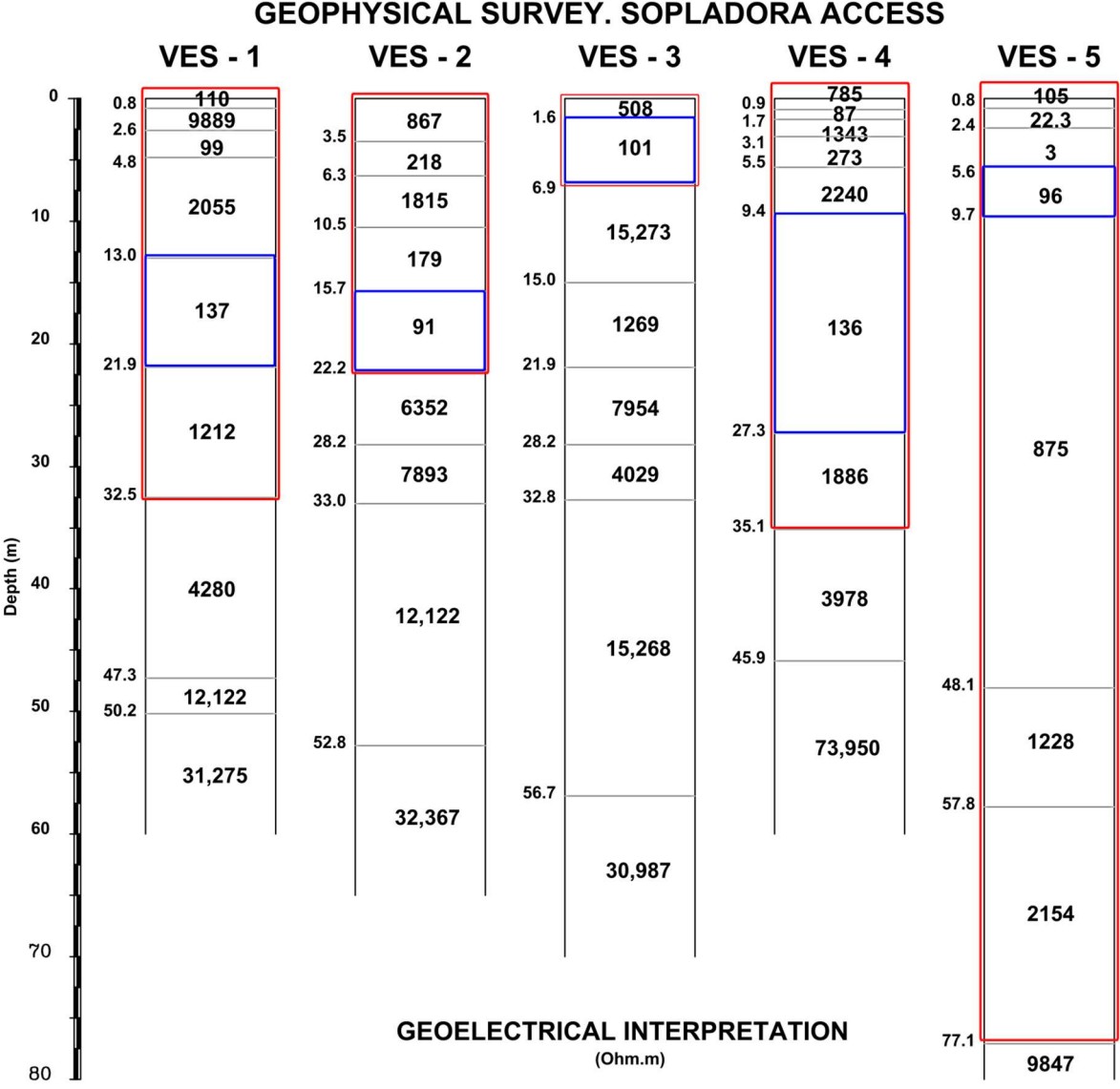

**Figure 7.** Results obtained in the interpretation of VES surveys (in Ohm.m). A red rectangle marks the defined sediment layers over the basement, and the blue rectangles show the water saturation levels.

## 6. Discussion

### 6.1. Data Integration and Calculation

The information obtained in the seismic refraction and passive MASW surveys, together with the interpretation of the VES surveys, has defined a zone of low compaction surface materials with velocities below 1000 m.s$^{-1}$ ($V_p$) and 600 m.s$^{-1}$ ($V_s$), which are correlated with resistivities lower than 1500 Ohm.m. These surface materials would correspond to vegetative and transported soils and colluvial ones with a high percentage of coarse material (pebbles and blocks) embedded in a clay-silt matrix. The geophysical levels below these materials' present characteristics, both in seismic and electrical techniques, of compact to very compact materials, which would correspond to a competent and cemented substrate.

From this considered model, the correlation between the fundamental frequencies $f_o$, obtained at the control points in the HVSR tests and the thicknesses of surface materials (sediments) in said areas obtained from the data of the SEV tests, has been proposed and executed at the same points.

Therefore, the results of the thickness of the sedimentary materials over the metamorphized rocky substrate are presented in Table 3, together with the values obtained in the

tests of the HVSR technique. The depths shown for each of the five VES surveys carried out also correspond to the values of the position of said basement in the refraction and MASW-type tests of the seismic method.

**Table 3.** Relationship between depth values obtained in VES and corresponding HVSR frequencies.

| VES Number | Rock Depth in VES Surveys (m) | Corresponding HVSR Survey | $f_o$ (Hz) |
|---|---|---|---|
| 1 | 21.9 | S1 | 3.21 |
| 2 | 22.2 | S2 | 2.61 |
| 3 | 32.8 | S3 | 0.70 |
| 4 | 35.1 | S4 | 0.49 |
| 5 | 57.8 | S5 | 0.25 |

This relationship is essential to the whole area to establish the empirical correlation concerning the fundamental frequencies obtained in HVSR surveys, which respond to Equation (1). Figure 8 shows the graphic correlation established between both groups of data in Table 3 and the value of the adjustment of the curve. In this case, said value ($R^2$) has been established in 0.932, a result that can be considered a good correlation value, even using the few available data, with which we can consider an estimation error of 10% in subsequent calculations.

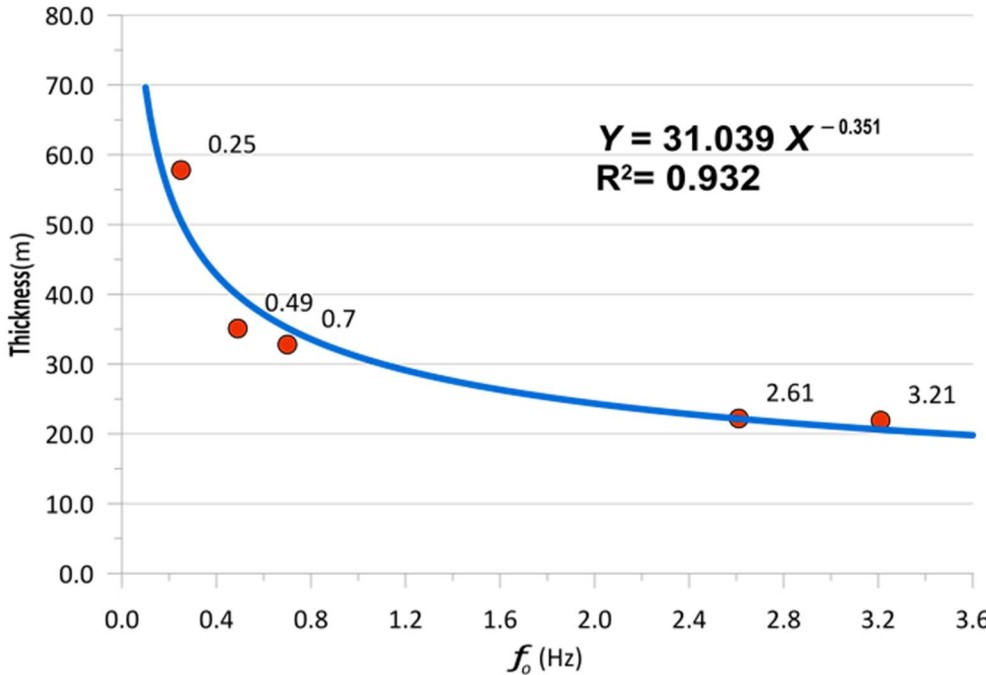

**Figure 8.** Empirical correlation established between VES (sediment thickness) and HVSR (frequencies) surveys.

Therefore, Equation (1) would be transformed according to the following values:

$$H = 31.039 \ f_o^{\ -0.351} \tag{3}$$

If we consider that $H$ is the value of the thickness of surface sediments expressed in meters, from that Equation (3), it can be calculated for each point of the HVSR test station measured and based on the value obtained from the fundamental frequency ($f_o$, in hertz), the value of the thickness of the sediments found over the rocky substratum.

Table 4 presents the calculations established for each of the measured HVSR station points, including the five corresponding to the control points (in the position of the VES

tests and indicated with the prefix S), applying the formula of Equation (3). To calculate the $K_g$ value, also shown in Table 4, the results obtained in Table 2 were used and applied Equation (2). As [46] considers, the values over 5 to 10 in $K_g$ value (it depends on area and materials conditions) are prone to show instability or are capable of showing it. In this case, it can be correlated with a susceptibility to the present capability of sliding or a movement downhill, so that they can mark areas of potential movements in a landslide.

**Table 4.** Results obtained for sediment thickness for all HVSR test points from Equation (3), and $K_g$ dimensionless values (using Table 4 values in Equation (2)).

| HVSR Point | $f_o$ (Hz) | Thickness (m) | $K_g$ |
|:---:|:---:|:---:|:---:|
| S1 | 3.21 | 20.61 | 1.75 |
| S2 | 2.61 | 22.16 | 1.61 |
| S3 | 0.70 | 35.18 | 31.56 |
| S4 | 0.49 | 39.87 | 23.87 |
| S5 | 0.25 | 50.49 | 61.15 |
| 6 | 1.07 | 30.31 | 16.41 |
| 7 | 3.66 | 19.68 | 1.42 |
| 8 | 57.67 | 7.48 | 2.78 |
| 9 | 0.34 | 45.33 | 123.88 |
| 10 | 3.28 | 20.46 | 1.95 |
| 11 | 2.42 | 22.76 | 1.49 |
| 12 | 3.76 | 19.50 | 0.63 |
| 13 | 0.73 | 34.66 | 13.85 |
| 14 | 0.60 | 37.13 | 14.90 |
| 15 | 9.37 | 14.15 | 0.58 |
| 16 | 1.04 | 30.61 | 8.60 |
| 17 | 4.81 | 17.88 | 0.85 |
| 18 | 2.95 | 21.23 | 1.54 |
| 19 | 0.71 | 35.00 | 16.47 |
| 20 | 0.91 | 32.08 | 11.39 |
| 21 | 25.30 | 9.99 | 0.22 |
| 22 | 1.18 | 29.29 | 8.52 |

The methodology proposed in this research has made it possible to establish the thickness of less compact materials (soft sediments not compacted o cemented), not only from direct investigations such as VES, but also in refraction or MASW seismic profiles. Therefore, the relationship established in Equation (3) defined the values of said thicknesses under each point of the HVSR point station.

The use of relationships between the thicknesses obtained from VES surveys with HVSR tests has very few publications references. Only [21] uses this correlation between sediment thickness investigation and VES-type tests, whereas other authors, such as [47], use correlations obtained from electrical tomography tests but also combine and use results from mechanical drilling. In both cases, they concluded that the application of the geoelectric method techniques in combination with the HVSR measurements offer reliable results in determining the thickness of surface sediments.

From the results obtained and presented in Table 4, these have been represented and analyzed by constructing a map of isolines of thickness values (isopach) that is presented in Figure 9. A special shape is located below HS5 and H9 points, which is the existence of a zone of great thickness (more than 55 m) in this central area and in relationship to the position of the greatest size of movement (just where the reinforced earth wall of Figure 2 has been built).

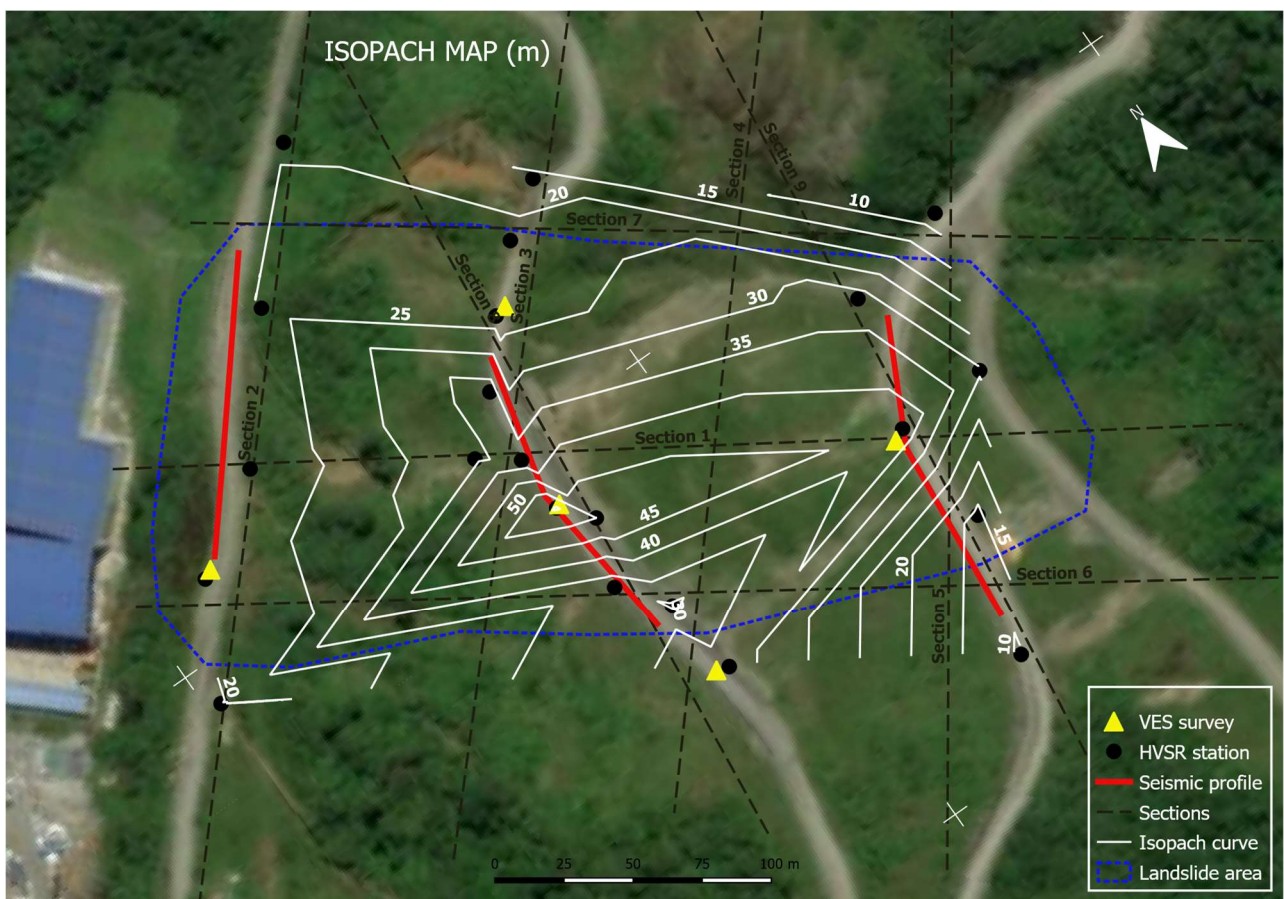

**Figure 9.** Isopach map (sediment thickness) traced with contour lines every 5 m thick (white lines) are represented with geological section positions (dashed black lines). White crosses refer to the coordinate points (Modified from [22]).

To complete the analysis, it was drawn over the isopach map, across several sections. Figure 9 shows the position of the sections that have been made to analyze these results obtained (black dashed lines). Figure 10 shows the longitudinal sections of the landslide (according to the direction of movement), and variations can be observed in the bottom and the position of the rocky substrate (Section 6) with a section (central Section 1) in which a typical distribution of a landslide with a thickening of the materials in the lower part caused by deposition.

In said longitudinal sections, it is also possible to observe the presence of a projection in the area of the lowest elevation (a peak from the basement), which may be one of the factors preventing the whole slide of all mass over the basement from continuing towards lower elevations of the terrain, which could be observed in the field. Lastly, in Section 7, traced by the zone where less movement is observed and the rock outcrops at some nearby point, the thickness of the sediment is less than 15 m.

The landslide surface traced in this section from the available data seems to respond to a translational landslide, or possible typology, since the thickness of moving materials along said section is maintained. However, the landslide can also have a rotational component or be considered a rotational –translational combination. In addition, by analyzing the morphology suggested by this research, it could be that the sliding area may respond to an accumulation of previous landslides (paleo-slides), as observed in nearby zones [44].

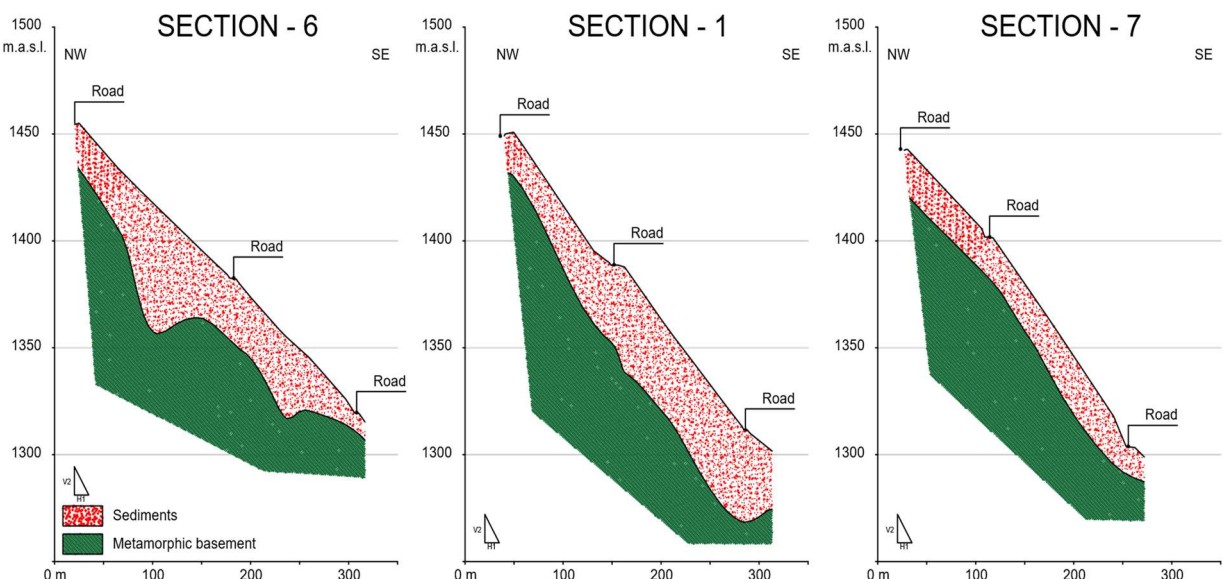

**Figure 10.** Simplified longitudinal sections of the investigated area are displayed, showing the landslide rupture surface position. Organized from south to north (left to right), the mobilized materials (soft sediments) over the basement are exhibited (Displaying scale 1H:2V).

A second depressed zone would be the outline of the shape of the landslide, in its central area, with a significant depression in the position of test H9 (the test that is also present in the longitudinal section) and with slopes to the north and south that present a steep slope.

The raised area that is observed more clearly in the cross section also seems to be evident, more gently in the central area of the longitudinal section, approximately where tests HS5 and H9 have been carried out (see Figures 4 and 9 for reference).

Four cross sections of the landslide (considering the movement direction) were traced on the area, and two others with the oblique north to south direction (see Figure 9 to identify its position). Figure 11 shows, from left to right, upper sections to lower ones. They show an important deepening at the center of the landslide area, and at Section 3 (over the intermediate position of the access road), a V-shaped area deeper than the other ones can be seen. The same deepening zone can be observed in oblique sections (see Figure 12) where the V-shaped sediment geometry is wider, but in all cross-sections, one of the flanks of that deepening area is sharply steep.

The possible presence of a fault structure in the basement, corroborated by low values in resistivity observed in VES-5, can be considered. Now, this depressed area can correspond to a ravine or temporary torrential channel (marked by the low values in sediments resistivity).

The cross-Sections 2, 3 and 5 obtained are concordant with the seismic refraction profiles interpreted (considering that seismic profiles were applied in different directions).

The graphical distribution of the value of the Vulnerability Index ($K_g$) is presented in Figure 13. It shows peak values in the area around VES-5 and H9 survey points and coincides with the greatest movement and the area of action carried out by the company E.P. CELEC with the construction of the reinforced earth wall. In addition, in this area, according to communication from the company's technicians, the movement continues to occur today.

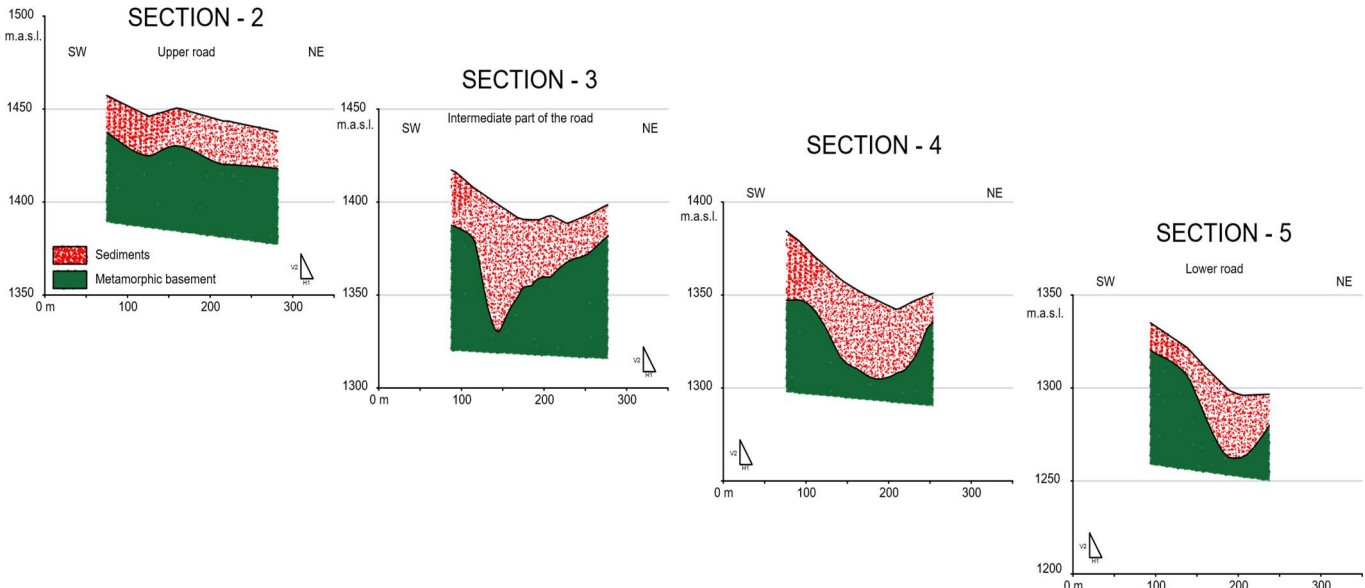

**Figure 11.** Cross sections from the upper (**left**) to the lower zone (**right**) showing sliding sediments over the basement (Drawing Scale relation: 1H:2V, keeping vertical relation of elevation and Figure 10 scale).

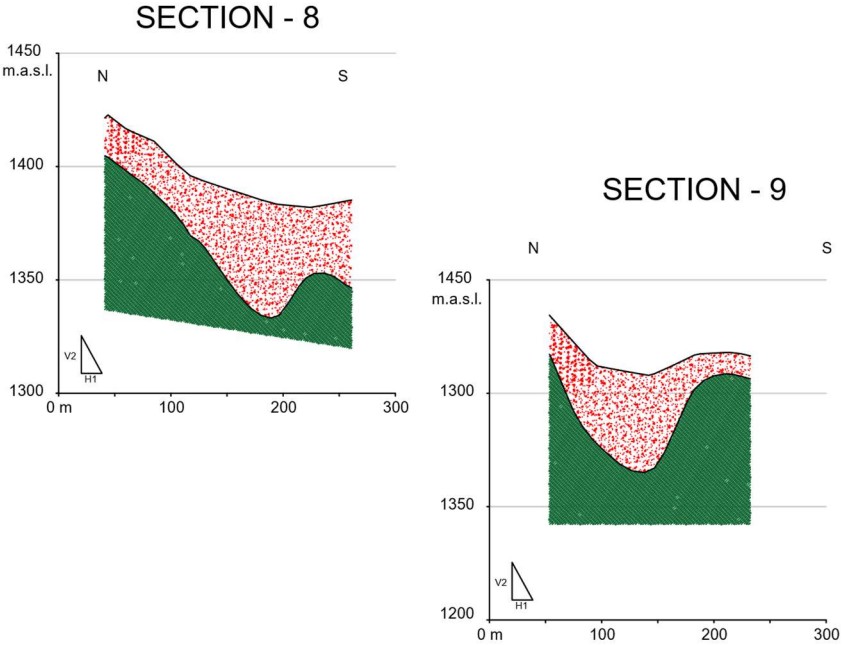

**Figure 12.** Oblique sections through the north–south direction represent the sliding sediments (in movement) over the basement where the rupture surface geometry is delineated (Drawing Scale relationship: 1H:2V, keeping the vertical relationship of elevation and Figure 10 scale).

This parameter is related to the ground shear strain or effective stress [46] and can be correlated with the potential ability of materials to move (sliding susceptibility); therefore, it can be recognized as a tool for assessing said capacity in landslides. In this study, in Figure 13, extremely high values appearing under the actual moving area (remarked by a dark blue continuous-line oval) and high values towards the upper area from the last one described (remarked by a dark blue dashed-line oval) can be observed. The rest of the points can be considered as low capabilities concerning sliding or as currently stable.

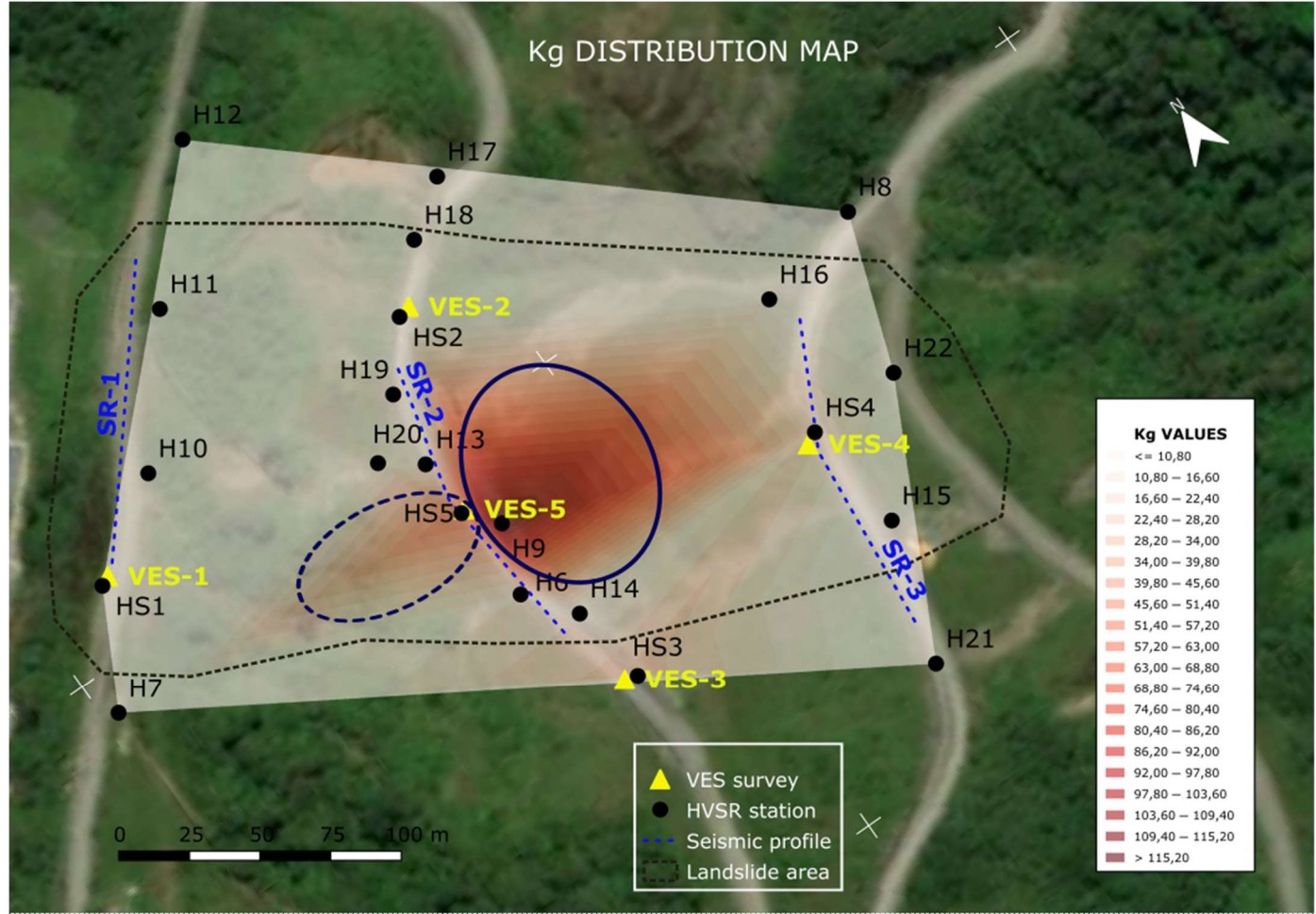

**Figure 13.** A distribution map of the $K_g$ values was represented in the investigation area. The maximums values coincide with the area presenting the most significant instability now (marked with dark blue ovals). White crosses refer to the coordinate points (Modified from [22]).

### 6.2. Data Correlation and Reliability Discussion

The results obtained in this landslide area are related to geophysical surveys alone, and no other direct data (as perforation bore-holes) is provided as verification. Using geophysical data, the indirect surveys considered could be an early investigation tool to delineate the characteristics of a sliding mass. It must be related to accurate direct information, but combining methods and techniques can improve the geophysical models [3–6].

In this area, the VES information shows a clear separation between shallow materials and the rocky substrate, with resistivity changing values up to four times (>5000 Ohm.m for the rocky substrate). Therefore, the obtained models could be used as a definition of the thickness of the soft material (sedimentary) and altered one (eluvium) with some grade of accuracy, thus defining the surface rupture of the landslide. In the metamorphic studied area, the materials overlaying the rocky substrate are those involved in landslides [44].

Moreover, seismic surveys provide similar distribution in shallow sediment layers using seismic velocity parameters, i.e., they have corroborated the VES geoelectrical interpretation. Furthermore, the models offer a contrast in impedance values between the sediments and the basement that is up to 2.2, which is a piece of important information in applying the HVSR two-layer model [10,15–17,21].

The use of few contrast points could be a limitation too. Of course, the more that can be used could help to obtain better accuracy. Nevertheless, the most important thing is to get a variety of thicknesses to adjust the empirical curve (Equations (1) and (3)), as can be

seen in other investigations (related in [20,21]). That accuracy and reliability in the final results are related to the precision of geophysical models that were employed [21,47].

The *a* and *b* factors in Equation (3) differed from those obtained by other authors (applied in basin geological areas) such as [18] or [47]. Considering [33,48], they indicated that *a* factor is related to local geological characteristics (ground materials, impedance contrast, and humidity). In contrast, the *b* factor is related to sediment thickness and the geometrical shape of the basement. The obtained factors here are close to those obtained in [20], an investigation also made in Ecuador, where geological conditions are more similar to the present one.

## 7. Conclusions

In this work, a landslide with an approximate area of 65,000 m$^2$ in Guarumales (Ecuador) has been analyzed. One and two-dimensional type geophysical surveys of seismic and electrical methods have been jointly implemented. Specifically, five VES surveys (used as control points) were performed at the same position as the other five HVSR single-station measures. Three seismic profiles (refraction and MASW types) were carried out at different elevations in the landslide area to complement the geophysical surveys. These techniques were used to confirm and perform the final analysis model in a total of 22 HVSR point measures.

At the parametric control points, the thicknesses of the sliding materials (those that exhibit shallow lower resistivity and seismic velocity values) were defined from those of the compact substrate (considered fixed, with higher geophysical values). The empirical relationship obtained from VES points results related to soft sediments thickness correlated with the natural frequencies values of the ground $f_o$ (obtained from HVSR surveys executed at control points). The empirical Equation (3) obtained has an *a* factor of 31.039 and a *b* factor of $-0.351$, with an adjustment error of R$^2$ = 0.932, which is considered a good correlation value.

From this Equation (3), the sediment thicknesses have been calculated and contoured on an isopach map using the values obtained at the different points from the measured HVSR stations. On those map results, several sections of the terrain can be executed (longitudinal, oblique, and transversal ones, considering the direction of the material slide). These sections have shown the existence of a deeper area V-shaped with steep flanks. This deep sedimentation could be related to a possible fault structure affecting the metamorphic basement.

The potential slip rupture surface, with a translational or roto-translational-type shape, has been proven to be related to the separation between soft materials (shallow sediments) layered over the basement (considered as fixed material).

Complementarily, the $K_g$ vulnerability index value analysis related to ground shear strain shows two prone areas to be the potential to continue the movement. These areas are in the exact location where emergency constructions were made to keep the integrity of the affected road.

The methodology in combining VES-type and HVSR-type surveys in investigating surface sediments showing in an impedance contrast (over 2.2) in defining sedimentary sliding material overlying the fixed basement has been verified. This methodology is easy, quick to apply and interpret, and has low economic costs. It can be transferred to other areas with access to complicated drilling machinery and as a tool prior to more elaborate or expensive economic surveys, considering the conditions (definition of contrast values in geophysical parameters).

The use of VES and HVSR techniques, and the methodology developed in this research, can be an investigation and decision-making tool in the monitoring and instrumentation of slippery zones caused by the speed of execution of the surveys and their economy.

Based on these results, it would be convenient to establish a complementary research campaign through bore-hole drilling with core recovery to establish the conditions of the materials and verify what was exposed in this investigation.

**Author Contributions:** Conceptualization, O.A.-P. and F.J.T.; methodology, O.A.-P.; software, O.A.-P.; validation, O.A.-P., J.G.-R. and A.G.; formal analysis, O.A.-P. and F.J.T.; investigation, O.A.-P. and F.J.T.; resources, O.A.-P. and F.J.T.; data curation, O.A.-P. and A.G.; writing—original draft preparation, O.A.-P.; writing—review and editing, J.G.-R. and F.J.T.; visualization, O.A.-P. and A.G.; supervision, J.G.-R. and F.J.T. All authors have read and agreed to the published version of the manuscript.

**Funding:** This research received no external funding.

**Institutional Review Board Statement:** Not applicable.

**Informed Consent Statement:** Not applicable.

**Data Availability Statement:** All data and processing results are under request. They will be available to anyone when they submit a request to the corresponding author.

**Acknowledgments:** This research work has been possible thanks to the collaboration of Ing. Iván Javier Hidrobo Montoya, manager of Unidad de Negocio Hidropaute, Corporación Eléctrica del Ecuador-CELEC E.P., and Ing. Ximena Robles, who allowed the use of the data to conduct this investigation. Part of the VES surveys data and seismic refraction profiles data presented in this document, as well as the topographic mapping handled, are the property of CELEC E.P.-SOUTH.

**Conflicts of Interest:** The authors declare no conflict of interest.

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
