# Peer review of "Early Investigation of a Landslide Sliding Surface by HVSR and VES Geophysical Techniques Combined, a Case Study in Guarumales (Ecuador)"

_applsci, doi:10.3390/app13021023_

Round 1
Reviewer 1 Report
The paper seemed me very interesting and it could be a good contribution about knowledge of landslides in Ecuador. Authors have carried out a detailed analysis and interpretation of a landslide case study in Ecuador by using geophysical techniques. I think manuscript should be accepted, although the authors should make major changes to the text.
Firstly, key words are correct. However, I would put Guarumales (Ecuador) the first key word.
The abstract is good and summarizes successfully the content of the paper. However, I think acronyms of SEV and HVSR (Line 21) must have their full names. Also, the phrase “There is also evidence of surface” (Line 27) does not understand. ¿What do you mean: surficial water evidence?
The Introduction section is correctly structured and it is easy to read. Nevertheless, there are some paragraphs very repetitive, in places that do not correspond or are badly described:
Paragraph of Lines 56 – 59 is very long. I would list the different geophysical techniques with 1), 2), 3). On the other hand, it is necessary to add a lot of references that explain the use of these techniques in the study of landslides.
Paragraphs of Lines 75 – 78 and 89-92 must be rewritten because they are not understood.
Lines 137 – 143 are not in the correct sub-section. I think they should be in sub-section background and scope.
Finally, I think paragraph between Lines 154 and 163 is repetitive with respect to introduction text. I think these lines could be in Lines 56-69.
As for Geophysical research and applied methodology section, the text and structure are fine. However, there are some points that need be improved.
Lines 299 and 301 are main aims of this work. So, I think they would be in Introduction (Lines 75-78).
Acronym VES in Line 311-312 has been already mentioned before. Then, it is not necessary again. Please, review all acronyms along the text.
Although there are an adequate number of figures in the manuscript, in my personal opinion, a flow diagram that displays all techniques applied and steps followed in this research can be very useful to comprehend better this article.
Regarding Results, they are generally well described and structured. In this section, few small points should be considered: 1) In my personal opinion, m/s should be changed by m s-1, 2) to be always homogeneous with decimals (one or two numbers and always with points and not commas) and 3) Figures and tables are cited a few times in the text. For example, Figure 5 is only cited two times. In this case, I would cite it at the end of lines 396, 402 and 405. Please, this criterion should be applied to all figures and tables.
Discussion is correct. However, this section could be improved. I think limitations found during the research could be mentioned with respect the techniques used. Also, you could indicate the necessity of using another techniques, such as UAV photogrammetry or A-DInSAR methods, because they can be useful to improve the results and interpretations of this research. These techniques can estimate velocity rates in millimetric scale along several years. In addition, I think you should say something about rainfall and moisture role in landslides and their possible analysis in the future, because they are usually triggering factors of these terrain movements.
With regard to the conclusions, I think it is necessary to add an introductory phrase as: "In this work a landslide with an approximate area of X m2 in Guarumales (Ecuador) has been analysed."
Also, it is necessary indicate some numerical results like average and maximum depths and vulnerability index obtained.
Figures 1, 3, 4, 5, 9 and 13 are not appropriate for the article because resolution and quality are very poor. So, I indicate possible improvements for them below:
Figure 1 and Figure 3 have got low resolution. In the case of the first Figure, I miss a DEM and/or level curves that show the local relief in the study area. In Figure 3 legend, text and scale should be changed and improved because they are very blurred.
Figure 4 has got a very low resolution. It should be changed by a better orthophoto or a high resolution DEM. Moreover, if you use orthophotos, you must always indicate the exact date of the images. Also, scale is very little and it is difficult to seeing.
In the case of Figures 5 and 6, the font size should be increased.
Figures 9 and 13 need similar changes than Figure 4: resolution and scale should be improved. Another possibility is to add a DEM with an adequate cell resolution.
Finally, the text should be checked and reviewed carefully. The following cases are some examples:
Line 37: after (Ecuador) delete ;.
Line 145: remove bold letters.
Line 150: Reference source not found.
Lines 177 – 178: The phrase “there is where the movement is prone to continue but at a low tax of displacement” cannot be comprehended. An option could be: “In this location, the movement presents low rates of movement.”
Line 208: add coma between 38 and 39.
Line 292: I disagree in using vergence in this line because this concept is more appropriate in Structural geology and, more specifically, in folds. So, this line should be changed.
Line 628: the phrase “have been applied combined” could be changed by “have been jointly implemented”.
Table 3 and Table 4 present decimals with commas. Please, change them by points.
Reviewer 2 Report
The manuscript titled “Early investigation of a landslide sliding surface by HVSR and VES geophysical techniques combined. A case study in Guarumales (Ecuador)” proposes a combination of the HVSR technique and VES geoelectrical surveys to investigate the characteristics of landslide. The authors aim to provide an overview of the sedimentary material overlaying compact material and correlate it with the rupture surface’s position. I agree that such method can be significant in landslide studies. However, the manuscript is not logically organized, and their methodology is not clearly described. Unfortunately, I decide to reject this manuscript. An intensive revision is required before we can consider proceeding with this manuscript.
First, the background, scope, and methodology of this paper are not logically organized. Readers may feel puzzled with the structure of the statement, which jumps between the geological background, HVSR surveys, and VES surveys continuously. It lacks detailed introduction of the VES technique, which is important in defining the thickness of the altered materials. It is not enough to simply mention that “The VES surveys have provided a distribution of materials that have been summarized in Figure 7”.
Second, as the authors indicated, if the soil or sediments have lateral changes in composition or compaction, there could be irregularities in the definition of the depth of the basement. Please explain how to solve the problem when conducting the HVSR survey. In addition, please discuss the effects of underground water on the HVSR surveys.
There are also some problems with the format of this submitted manuscript. For example, in Line 344 it is written “Section 6.1”. The numbers of Lines 386, 411, 434, and 466 are not correct, either. There is an error with the citation in Line 150. Please carefully check the format before re-submitting the manuscript.
Last but not least, the quality of figures need improvement. For example, do not use comma to represent the decimal point in Figure 8. Use dot instead. Please also add a color bar to Figure 13.
Round 2
Reviewer 2 Report
I thank the authors’ efforts in revising the manuscript. Compared to the last version, it is more logically organized and the introduction of VES technique was more clearly described. I am satisfied with the response to my comment concerning the irregularities in the definition of the depth of the basement. However, the new problem is that the Introduction Section became quite long, which makes this manuscript verbose. The second section “Geological knowledge” can also be regarded as a part of the long introduction. Therefore, I strongly recommend that the authors should reshape the manuscript. Please shorten the introduction and remove some contents (e.g., geological knowledge” to supplementary material.
Another problem of this manuscript is that it lacks the comparison with traditional method based on their results. It only says that “the application of the geoelectric method techniques in combination with the HVSR measurements offers reliable results…”. Here more discussions on the reliability are expected. Otherwise, it is difficult to find out the scientific contribution of this study.
Because of these two problems, I recommend a major revision before we can proceed with this manuscript.
Round 3
Reviewer 2 Report
I thank the authors’ efforts in reshaping the Introduction Section and improving the Discussion Section. The revised version of the manuscript titled “Early investigation of a landslide sliding surface by HVSR and VES geophysical techniques combined. A case study in Guarumales (Ecuador)” is now suitable for publication in Applied Sciences. I am glad to accept this paper.